# Two-dimensional metallic tantalum disulfide as a hydrogen evolution catalyst

Jianping Shi[1,2], Xina Wang [3], Shuai Zhang[4], Lingfeng Xiao[3], Yahuan Huan[1,2], Yue Gong[5], Zhepeng Zhang[2], Yuanchang Li[4], Xiebo Zhou[1,2], Min Hong[1,2], Qiyi Fang[1,2], Qing Zhang [1], Xinfeng Liu[4], Lin Gu[5,6,7], Zhongfan Liu[1] & Yanfeng Zhang[1,2]

Two-dimensional metallic transition metal dichalcogenides are emerging as prototypes for uncovering fundamental physical phenomena, such as superconductivity and charge-density waves, as well as for engineering-related applications. However, the batch production of such envisioned transition metal dichalcogenides remains challenging, which has hindered the aforementioned explorations. Herein, we fabricate thickness-tunable tantalum disulfide flakes and centimetre-sized ultrathin films on an electrode material of gold foil via a facile chemical vapour deposition route. Through temperature-dependent Raman characterization, we observe the transition from nearly commensurate to commensurate charge-density wave phases with our ultrathin tantalum disulfide flakes. We have obtained high hydrogen evolution reaction efficiency with the as-grown tantalum disulfide flakes directly synthesized on gold foils comparable to traditional platinum catalysts. This work could promote further efforts for exploring new efficient catalysts in the large materials family of metallic transition metal dichalcogenides, as well as exploiting their applications towards more versatile applications.

[1] Department of Materials Science and Engineering, College of Engineering, Peking University, Beijing 100871, China. [2] Center for Nanochemistry (CNC), Beijing Science and Engineering Center for Nanocarbons, Beijing National Laboratory for Molecular Sciences, College of Chemistry and Molecular Engineering, Peking University, Beijing 100871, China. [3] Hubei Collaborative Innovation Center for Advanced Organic Chemical Materials, Faculty of Physics and Electronic Technology, Hubei University, Wuhan 430062, China. [4] Division of Nanophotonics, CAS Key Laboratory of Standardization and Measurement for Nanotechnology, CAS Center for Excellence in Nanoscience, National Center for Nanoscience and Technology, Beijing 100190, China. [5] Beijing National Laboratory for Condensed Matter Physics, Institute of Physics, Chinese Academy of Sciences, Beijing 100190, China. [6] Collaborative Innovation Center of Quantum Matter, Beijing 100190, China. [7] School of Physical Sciences, University of Chinese Academy of Sciences, Beijing 100190, China. Correspondence and requests for materials should be addressed to Y.Z. (email: yanfengzhang@pku.edu.cn)

Two-dimensional (2D) metallic transition metal dichalco-genides (MTMDCs) such as TiSe₂[1–5], NbSe₂[6–9], TaS₂[10–13], and TaSe₂[14–16], have kindled worldwide research interest due to their rich phase diagrams that include superconductivity, charge-density wave (CDW) and metal-insulator transitions. These intriguing properties are mainly attributed to their reduced dimensionality and the induced quantum confinement effect. Recently, such unique 2D systems have become appealing plat-forms for exploring the origin of superconductivity and CDW, longstanding puzzles in condensed matter physics[17–19]. For instance, the coexistence of CDW order and superconductivity has been unveiled in atomically thin TaS₂[11, 12, 20]. However, the TaS₂ samples reported in a majority of the publications were obtained by an exfoliation method[10–13, 20], which is time-con-suming, incompatible with batch production, and affords little control over thickness and domain size.

Chemical vapour deposition (CVD), compatible with common tool sets and scalable syntheses, has been regarded as a swift and effective route for growing semiconducting TMDCs (e.g. MoS₂[21–23], MoSe₂[24, 25], WS₂[26, 27], ReS₂[28, 29], etc.). Very recently, this approach has been extended to the synthesis of MTMDCs[30–32]. For example, Lou et al. synthesized metallic VS₂ single-crystal flakes on SiO₂/Si through an atmospheric pressure

CVD (APCVD) route, with thicknesses ranging from ~100 to ~1100 nm[30]. Subsequently, Zhang et al. optimized the growth conditions and obtained ~8-nm-thick VS₂ flakes[31]. Meanwhile, Liu et al. reported the APCVD growth of 1T-TaS₂ flakes on SiO₂/Si with a wide thickness range of 2~220 nm[32]. Nevertheless, large-area syntheses of full coverage or large-domain MTMDCs and identification of their possible applications are still works-in-progress.

Experimental and theoretical efforts have indicated that MoS₂ nanoparticles or nanosheets are potential electrocatalysts for the hydrogen evolution reaction (HER)[33–35], and metallic 1T-MoS₂ can be much more active than its semiconducting counterpart[36]. However, such 1T-MoS₂ is vulnerable to ambient conditions, and its direct synthesis through reactive alkyl lithium intercalation is difficult. It thus appears reasonable to seek phase-stable MTMDCs to replace 1T-MoS₂ for realizing efficient catalytic applications. Recent theoretical calculations have predicted the possibility of TaS₂ as an active and stable electrocatalyst[37]. From the experimental side, Chen et al. performed HER measurements of liquid-phase-exfoliated 1T-TaS₂, and reported an enhanced catalytic activity for atomic-scale-pore decorated TaS₂ (intro-duced via oxygen plasma treatment) over the conventional form[38]. However, the reported HER performance was still not

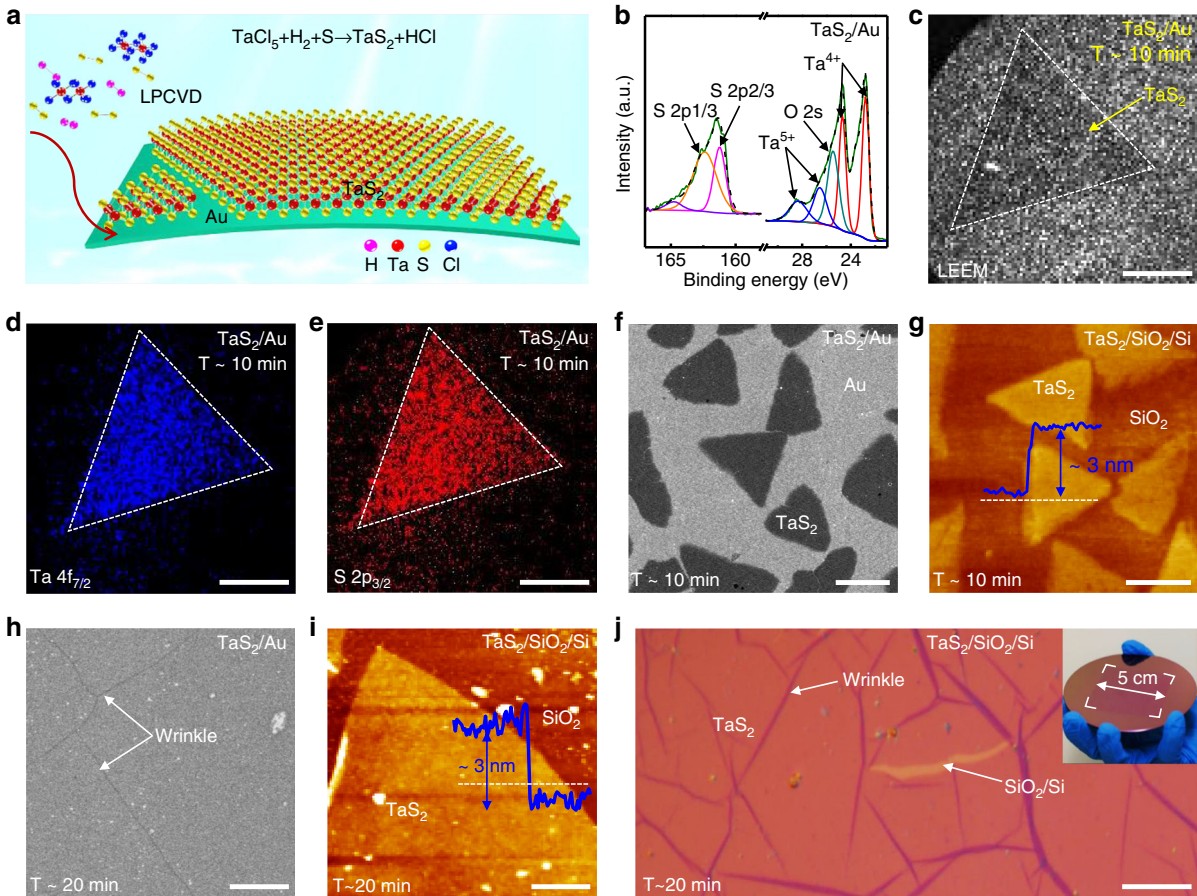

**Fig. 1** LPCVD synthesis of centimetre-sized uniform ultrathin 2H-TaS₂ films on Au foils. **a** Schematic illustration of the LPCVD growth process. **b** XPS peaks of Ta and S in as-grown 2H-TaS₂, respectively. **c–e** Synchrotron radiation-based LEEM and μ-XPS elemental mapping of Ta (4f₇/₂) and S (2p₃/₂) acquired on consecutive areas of 20 × 20 μm² (synthesized at ~750 °C for ~10 min under Ar/H₂ (~100/10 sccm) carrier gases), confirming the formation of near triangular TaS₂ flakes. **f** Corresponding SEM image of as-grown 2H-TaS₂ flakes on Au foils. **g** AFM image and corresponding height profile of transferred 2H-TaS₂ flakes on SiO₂/Si showing a nominal thickness of ~3 nm. **h** Large-area ultrathin 2H-TaS₂ film evolved on Au foils by further prolonging the growth time to ~20 min (with the other parameters keep identical to that of **c, f**). **i** AFM height image of a transferred film edge presenting the same thickness as the initially evolved flakes, as evidenced by the inset height profile analysis (~3 nm). **j** Large-area OM image indicating the centimetre-size uniformity of the transferred 2H-TaS₂ film on SiO₂/Si (synthesized at 750 °C for 20 min under Ar/H₂ (~100/10 sccm) carrier gases). Inset is the photograph of 2H-TaS₂ film on wafer-scale SiO₂/Si. Scales bars, 5 μm in **c–e**, 10 μm in **f, g**, 20 μm in **h**, 50 μm in **i** and 0.5 mm in **j**

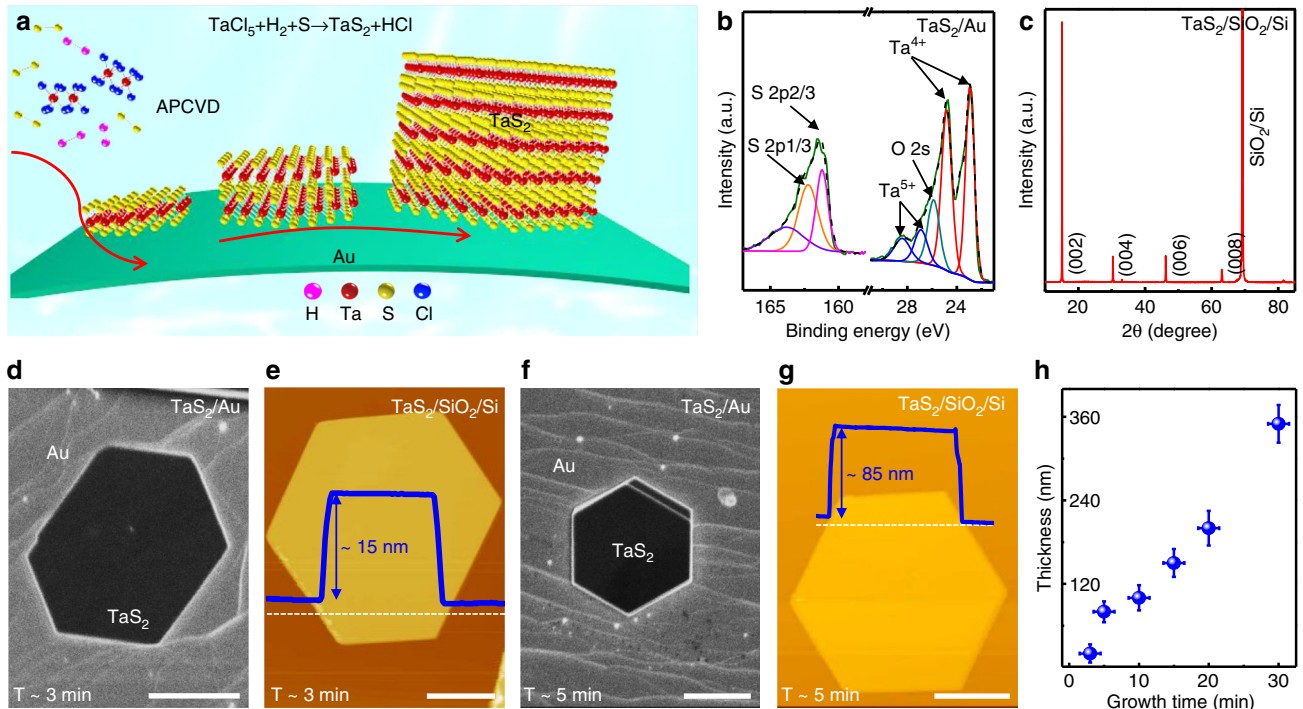

**Fig. 2** APCVD growth and characterization of 2H-TaS$_2$ hexagonal flakes. **a** Schematic illustration of the APCVD growth process. **b** XPS peaks for Ta and S confirming the formation of 2H-TaS$_2$ on Au foils. **c** XRD pattern of transferred 2H-TaS$_2$ on SiO$_2$/Si showing its 2H phase feature. **d–g** SEM and corresponding AFM images showing the tunable thicknesses of hexagonal 2H-TaS$_2$ domains by varying growth time from ~3 to ~5 min (synthesized at 750 °C under Ar/H$_2$ (~100/10 sccm) carrier gases), respectively. **h** Plot of the thickness of 2H-TaS$_2$ as a function of growth time. Error bars are defined as s.d. Scale bars, 4 μm in **d**, **e** and 10 μm in **f**, **g**

comparable with that of 1T-MoS$_2$[33, 34], presumably due to the slight oxidation of atomic-scale pores. Moreover, the use of a common glassy carbon working electrode possibly restricted the electron transfer from electrode to catalytically active sites, due to a weak interface interaction.

To tackle the above-mentioned issues, here we design low-pressure CVD (LPCVD) and APCVD routes for the direct syntheses of centimetre-sized uniform, ultrathin TaS$_2$ films and thickness-tunable TaS$_2$ flakes on a common electrode material of Au foil, respectively. This provides us with an opportunity to explore either fundamental physical phenomena or related applications associated with the dimensionality effect. In particular, the nearly commensurate CDW (NCCDW)/commensurate CDW (CCDW) phase transition is unambiguously demonstrated, suggesting that the crystalline quality of CVD-derived TaS$_2$ is comparable to the mechanically exfoliated material. More significantly, the as-grown metallic TaS$_2$ on Au foils displays high electrocatalytic activity for the HER. The internal reaction mechanism is revealed by a combination of experimental results and theoretical calculations.

## Results

**LPCVD synthesis of centimetre-sized ultrathin 2H-TaS$_2$ film.** TaS$_2$ thin films were successfully synthesized by a LPCVD route with solid TaCl$_5$ and S as precursors, as depicted in the schematic view in Fig. 1a. In contrast to previous work reporting the synthesis of TaS$_2$ flakes on SiO$_2$/Si[32], we selected Au foil as a substrates due to its chemical inertness towards S precursors, its catalytic activity in TMDCs growth, and more significantly, its compatibility with large-area growth and direct application in HER[23, 27]. X-ray photoemission spectroscopy (XPS) measurements were firstly carried out to determine the chemical composition of the as-grown samples (Fig. 1b, and Supplementary

Fig. 1). The obtained Ta 4f$_{7/2}$ (22.7 eV) and 4f$_{5/2}$ (24.7 eV) peaks are attributed to Ta$^{4+}$, while the S 2p$_{3/2}$ (162.1 eV) and 2p$_{1/2}$ (163.2 eV) peaks are assigned to S$^{2-}$, in agreement with the standard XPS data of TaS$_2$[32]. The Ta:S atomic ratio calculated from the XPS data is 1:2.08, approximating to the 1:2 stoichiometric ratio for bulk TaS$_2$. Notably, additional peaks at 28.4 and 26.9 eV are attributed to Ta$^{5+}$, in consideration of the oxidation susceptibility of metallic TaS$_2$.

Furthermore, synchrotron radiation-based low-energy electron microscopy (LEEM) and micro-beam XPS (μ-XPS) measurements were also performed directly on as-grown samples (Fig. 1c–e). Figure 1d, e reveal the spatial mapping of Ta (4f$_{7/2}$) and S (2p$_{3/2}$), respectively, from which the shape/location of TaS$_2$ flakes can be definitively distinguished. The uniform contrasts within the triangular domains indicate the relatively high crystal quality of the CVD-derived samples. The X-ray diffraction (XRD) pattern of TaS$_2$ confirms its 2H phase structure (Supplementary Fig. 2), which is different from the previous report (1T-TaS$_2$ synthesized on SiO$_2$/Si with an APCVD route[32]). This difference can be explained by the relatively low growth temperature, the slow cooling process, and the different substrate used in this work. Notably, the 2H-TaS$_2$ should deliver a higher electro-catalytic activity than that of 1T-TaS$_2$, as predicted by theoretical calculation[37].

Scanning electron microscopy (SEM) examinations were then performed to show the morphology and the domain size evolution of 2H-TaS$_2$ on Au foils under different growth times of ~5, ~10, and ~20 min (Fig. 1f, and Supplementary Fig. 3). The domain sizes were found to be variable from ~0.5 to ~20 μm. In particular, at the growth time of ~10 min, the edge length of the 2H-TaS$_2$ triangle was as large as ~20 μm, as shown in Fig. 1f. An apparent height of ~3 nm was determined from the atomic force microscopy (AFM) section-view analysis across the domain edge

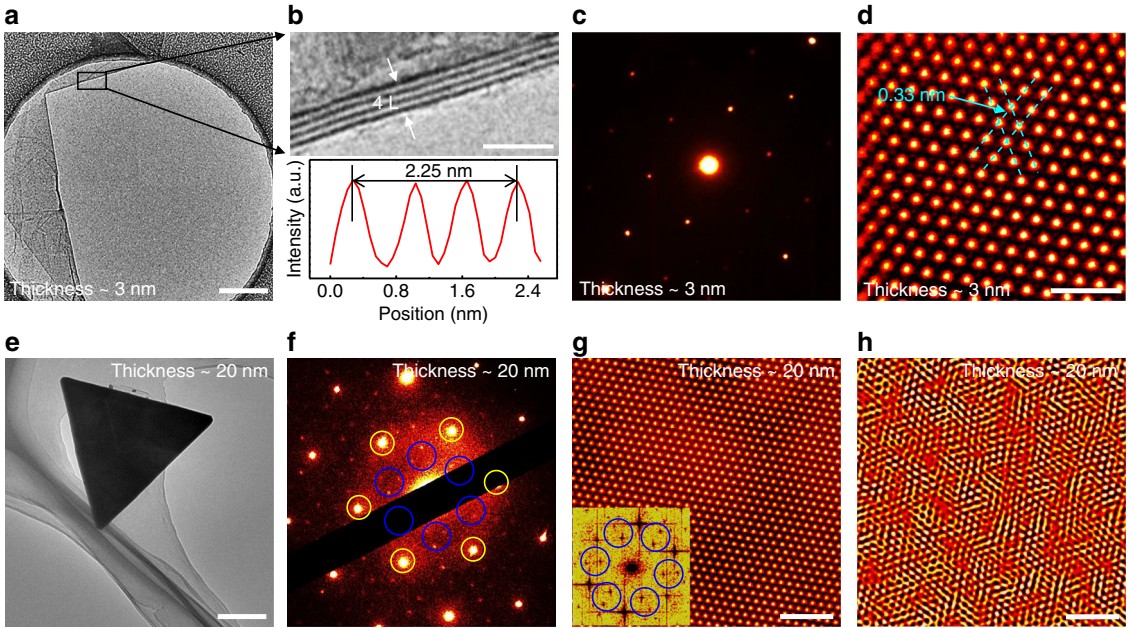

**Fig. 3** TEM characterization of the atomic structure of 2H-TaS₂. **a** Low-magnification TEM image of the LPCVD-derived 2H-TaS₂ film. **b** Magnified TEM image along the film edge in **a** showing its 4-layer feature. The bottom panel shows the corresponding line profile along the white arrow. **c** Corresponding SAED pattern captured from **a** within a 500 × 500 nm² area. **d** Atomic-resolution TEM image of the transferred sample. **e** Low-magnification TEM image of a triangle domain with the thickness of ~20 nm. **f** Corresponding SAED pattern captured from **e** within a 2 × 2 μm² area. **g** Atomic-resolution STEM-HAADF image showing the perfect atomic lattice. Inset is the corresponding FFT pattern. **h** Selective IFFT filtered image of NCCDW peaks in **g**) showing disordered periodic lattice distortion. Scale bars, 1 μm in **a**, **e**, 3 nm in **b**, **g**, **h** and 1 nm in **d**

of the transferred 2H-TaS₂ on SiO₂/Si (Fig. 1g). However, at a growth time of ~20 min, a full coverage 2H-TaS₂ film was obtained according to the uniform SEM contrast in Fig. 1h. The AFM image of the transferred 2H-TaS₂ in Fig. 1i reveals a layer thickness of ~3 nm, the same as that achieved with reduced growth time (Fig. 1g). The excellent thickness uniformity at the centimetre-size was further confirmed by a highly homogeneous optical microscopy (OM) image of the transferred 2H-TaS₂ on SiO₂/Si (Fig. 1j, and Supplementary Fig. 3). Note that, for other intermediate growth times, the thickness of the derived 2H-TaS₂ flakes maintained a similar value of ~3 nm. In this regard, we can infer that the current 2H-TaS₂ growth exhibits a 'magic' starting thickness of ~3 nm, i.e., individual islands evolved on the surface and then expanded with the increase of growth time, and finally merged together towards the formation of a complete layer. To the best of our knowledge, this is the first report about the synthesis of centimetre-sized uniform MTMDCs films.

It should be noted that the unique 2D growth feature is different from the self-limited surface growth of monolayer MoS₂ or WS₂ on Au foils[23, 27]. We ascribe this 'magic' growth behaviour to the dimerization of Ta along the c-axis direction, as similarly demonstrated in an analogue system of IrTe₂[39, 40]. A first-order structural transition from $1 \times 1 \times 1$ to $5 \times 1 \times 5$ was proposed for bulk IrTe₂, and this reconstructed structure possessed a five-times periodicity in both a and c directions of the crystal lattice. The existence of long-range ordering along the c direction (normal to the 2D plane) expressed an enhanced interplanar coupling with respect to traditional van der Waals coupled systems (i.e., MoS₂ and WS₂)[39, 40].

In order to confirm the aforementioned hypothesis, the growth time of 2H-TaS₂ was further prolonged to ~30 min. Some triangular 2H-TaS₂ flakes with the specific thickness of ~3 nm were then observed on the complete 3-nm-thick 2H-TaS₂ film (Supplementary Fig. 4). Such intriguing result strongly suggests the existence of a critical thickness for LPCVD synthesized 2H-

TaS₂ on Au foils. Notably, this growth behaviour has also been reported in the "electronic growth" of metallic overlayers on semiconductor substrates, wherein quantized electronic states (QWSs) were generated in the thin layers and determined the stability of the 2D thin films[41]. Briefly, centimetre-sized uniform, several-layer-thick 2H-TaS₂ films were successfully obtained, which should offer attractive playgrounds for exploring some fundamental physical issues, e.g. the interplay between CDW and superconductivity that has been disclosed in mechanically exfoliated layers[20].

**APCVD growth of 2H-TaS₂ flakes with tunable thickness.** Recent electrical transport measurements (temperature-dependence resistance) have revealed that the thickness of exfoliated TaS₂ flakes has a prominent influence on CCDW/NCCDW and NCCDW/incommensurate CDW phase transitions[11]. When the thickness was reduced to ~3 nm, such transitions suddenly vanished. In this regard, it is logical to synthesize high-quality, thickness-variable TaS₂ so as to explore this thickness-dependent phenomenon. APCVD has proven to be an effective method to grow semiconducting TMDCs with tailored thicknesses, due to the excess precursor feeding rate during the synthesis process[42, 43]. Motivated by this, we selected the APCVD approach to synthesize TaS₂ directly on Au foils, and the process is shown schematically in Fig. 2a. As a result, hexagonal TaS₂ flakes were successfully achieved as presented in Supplementary Fig. 5. The morphology variations from triangular to hexagonal shapes between APCVD- and LPCVD-derived TaS₂ are possibly attributed to the local changes of the Ta:S ratio of precursors, as previously demonstrated in MoS₂ growth[43]. If the Mo:S ratio was larger than 1:2, triangular MoS₂ domains were usually generated, however, when the Mo:S ratio was lower than 1:2, hexagonal MoS₂ flakes were usually evolved. Notably, the stoichiometric ratio of TaS₂ and its 2H phase structure were then confirmed by XPS and XRD measurements (Fig. 2b, c), the same as that of the

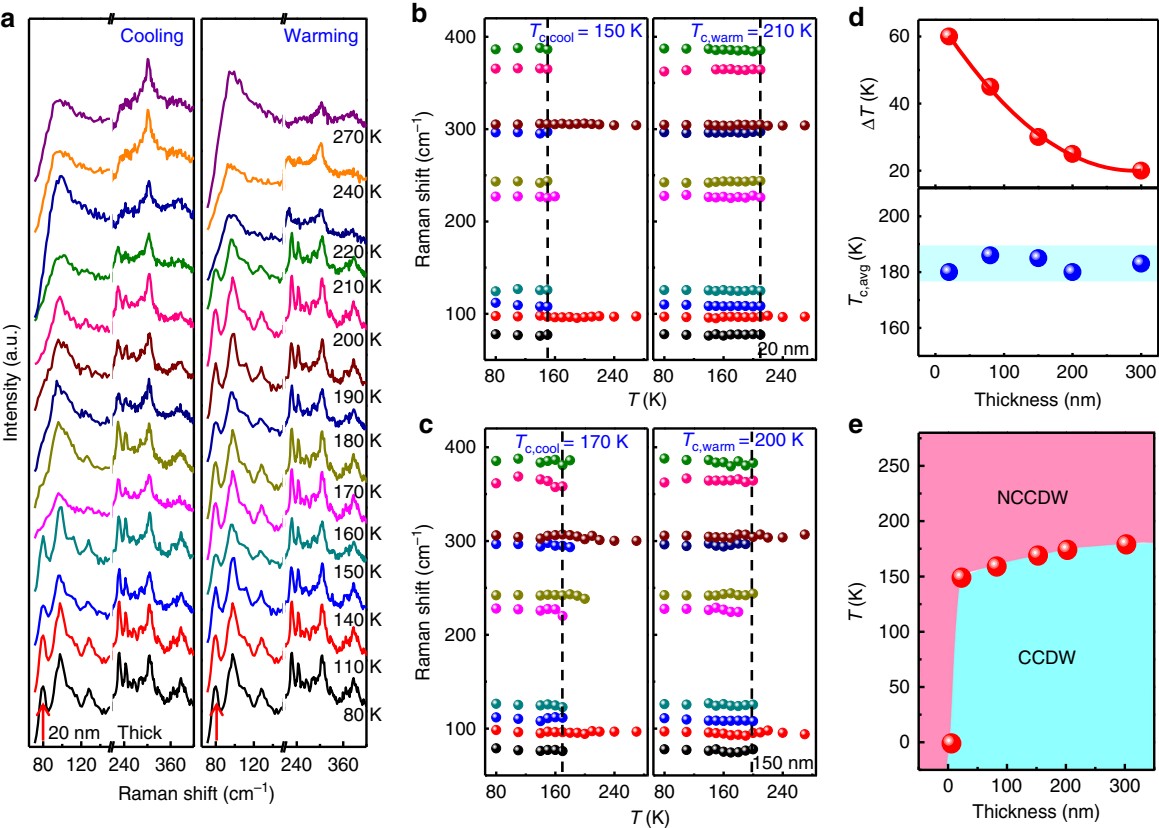

**Fig. 4** Temperature-dependent Raman characterization of 2H-TaS$_2$ flakes with different thicknesses. **a** Temperature-dependent Raman spectra of ~20 nm-thick 2H-TaS$_2$ captured from both cooling and warming processes. **b**, **c** Raman frequency plots of discernible peaks for ~20 nm-thick and ~150 nm-thick 2H-TaS$_2$ flakes with decreasing/increasing temperature, respectively. **d** Hysteresis (upper) and average transition temperature (lower) plotted as a function of sample thickness. **e** Thickness-temperature phase diagram of CVD-derived 2H-TaS$_2$ obtained from temperature-dependent Raman data. The red balls mark the boundary of NCCDW and CCDW phases. For simplicity, the phase boundary was recorded based on the cooling data

LPCVD synthesized films. The XPS measurement of transferred 2H-TaS$_2$ on SiO$_2$/Si was also performed to exclude the possible Au penetration into the 2H-TaS$_2$ layers during the CVD growth process (Supplementary Fig. 6).

Intriguingly, we found that upon increasing the growth time from ~3 to ~30 min, the edge length of the hexagonal 2H-TaS$_2$ flake can be tailored from ~5 to ~20 μm (Fig. 2d, f) and the thickness from ~15 to ~350 nm (Fig. 2e, g). This phenomenon highlights that the APCVD-synthesized 2H-TaS$_2$ on Au foils follows the Volmer-Weber (VW) growth mode, in contrast to the LPCVD growth obeying a Frank-van der Merwe (FM) mode. To provide further insight, the evolution of the flake thickness is plotted as a function of growth time (Fig. 2h), which clearly addresses the tunability of the thickness of 2H-TaS$_2$ by precisely varying the growth time. Altogether, high-quality, thickness-tunable, large-domain 2H-TaS$_2$ flakes can be synthesized on Au foils by an APCVD route.

**TEM characterization of 2H-TaS$_2$.** In order to obtain a thorough understanding of the crystal structure of the CVD-derived 2H-TaS$_2$, high-resolution transmission electron microscopy (HR-TEM) measurements were then performed on transferred samples. Figure 3a shows a low-magnification TEM image of the LPCVD-synthesized 2H-TaS$_2$, and the HR-TEM image captured from the film edge presents a layer thickness of 4, as well as an interlayer spacing of ~0.75 nm (Fig. 3b, and Supplementary Fig. 7). This data again confirms the existence of a 'magic' starting layer of 4, which we believe was mediated by the dimerization of Ta along the c-axis direction. Moreover, the corresponding

selected area electron diffraction (SAED) pattern in Fig. 3c reveals only one set of hexagonally arranged diffraction spots, strongly suggesting the single-crystalline nature of the 2H-TaS$_2$ domain. Atomic-resolution TEM image in Fig. 3d clearly displays a honeycomb structure with an interatomic distance of ~0.33 nm, as in good agreement with the documented lattice constant of TaS$_2$[20], further convincing the rather high crystalline quality of CVD-derived TaS$_2$.

Previous electrical transport measurements (temperature-dependent resistance) and TEM characterizations revealed that the CDW phase transitions of exfoliated TaS$_2$ were strongly suppressed at a reduced thickness of ~3 nm[11, 13]. In order to observe the CDW phase transition in 2H-TaS$_2$, the APCVD samples were characterized by TEM and spherical-aberration-corrected scanning transmission electron microscopy (STEM), a useful method to identify the CDW phases. The low-magnification TEM image in Fig. 3e shows a representative 2H-TaS$_2$ triangle of ~20 nm thick. Notably, the corresponding SAED pattern acquired at room temperature is rather complex (Fig. 3f). The bright spots (highlighted by yellow circles) correspond to the Bragg scattering from the triangular lattice of Ta atoms with a lattice constant $a = 0.33$ nm. This hexagonally arranged Bragg scattering pattern further confirms the single-crystalline feature of the 2H-TaS$_2$ triangle. However, the additional set of spots (indicated by blue circles surrounding the central beam) corresponds to the periodic lattice distortion (PLD) induced wave vectors, possibly due to the periodic atomic displacements of the NCCDW, as similarly reported for the exfoliated 1T-TaS$_2$[13].

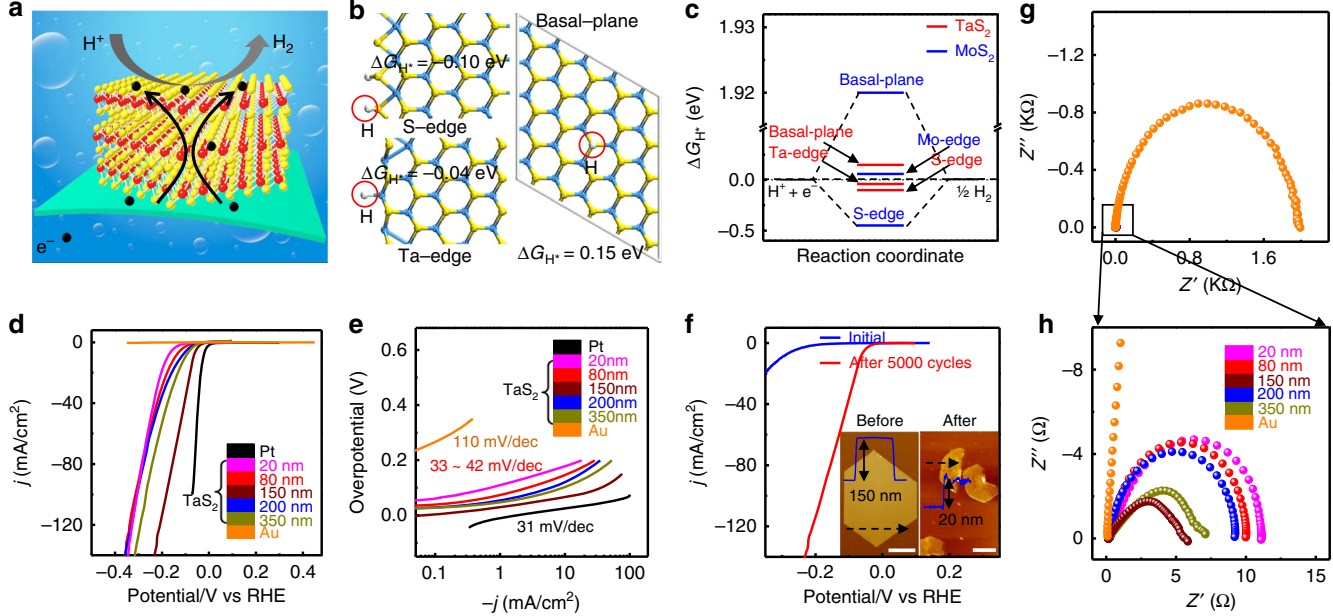

**Fig. 5** Electrocatalytic application of CVD synthesized 2H-TaS$_2$ in HER. **a** Schematic illustration of the HER process of 2H-TaS$_2$/Au foils. **b** Hydrogen adsorption energies at S-edge, Ta-edge, and basal-plane of 2H-TaS$_2$, respectively. Yellow, cyan, and grey balls represent S, Ta, and adsorbed H atoms, respectively. **c** $\Delta G_{H^*}$ diagram of different H adsorption states. **d** Polarization curves (*iR*-corrected) of as-grown 2H-TaS$_2$ with different thicknesses, Au foil, and commercial Pt. **e** Corresponding Tafel plots of the different samples in **d**. **f** Polarization curves (*iR*-corrected) of 2H-TaS$_2$ (~150 nm thick) before and after 5000 cycles. Inset is the corresponding AFM images. **g**, **h** Electrochemical impedance spectra of 2H-TaS$_2$ flakes with different thicknesses, as well as the Au foil substrate. Scale bars, 4 µm in (**f**, left) and 2 µm in (**f**, right)

In order to visualize the atomic-scale morphology and the NCCDW structure of 2H-TaS$_2$, atomic-resolution Z-contrast STEM-HAADF analysis was then carried out on the transferred sample. Figure 3g reveals a representative STEM-HAADF image obtained from the 2H-TaS$_2$ triangle in Fig. 3e. The Ta (bright spots) and S atoms can be clearly identified by their different contrasts. However, the S atoms are nearly invisible due to the large difference of the atomic number between Ta and S. Notably, the atomic arrangement obeys the 2H-phase atomic model with Ta atoms octahedrally coordinated by S atoms, as inferred by the intensity line profile in Supplementary Fig. 7. A further selective inverse fast Fourier transform (IFFT) filtered image of the diffraction spots (highlighted by blue circles) in Fig. 3g shows disordered PLD, suggesting the appearance of NCCDW phase state in TaS$_2$ (Fig. 3h). Briefly, the CVD-synthesized 2H-TaS$_2$ possesses comparable crystalline quality with that of mechanically exfoliated samples, which should allow more intensive investigations of physical phenomena such as CDW, superconductivity, and so on.

**Thickness-dependent CDW phase transitions of 2H-TaS$_2$.** The effect of dimensionality and interlayer coupling of TaS$_2$ on the CDW phase transition has aroused interest[11–13]. However, the existing electrical transport measurements (temperature-dependent resistance) and TEM analyses are usually time-consuming and complex, and inefficient in distinguishing the CDW phases arising from either bulk or sample surface. Raman spectroscopy has been established as an exquisitely sensitive and convenient technique to investigate both bulk and surface vibration modes of TMDCs[21, 22]. Very recently, temperature-dependent Raman spectroscopy has been utilized to determine the transition temperature of the CDW phase based on exfoliated 1T-TaS$_2$ flakes[44].

In our work, representative Raman spectra have been captured on a ~20 nm-thick 2H-TaS$_2$ flake upon cooling/warming processes, as shown in Fig. 4a. For the cooling process, a broad

Raman peak was clearly observed below 100 cm$^{-1}$ at > 150 K. In contrast, some fine peaks (indicated by a red arrow in Fig. 4a) are visible below 100 cm$^{-1}$ at < 150 K, suggesting that a NCCDW/CCDW phase transition takes place at ~150 K. Notably, a similar tendency to change is also recorded during the warming process but with a higher transition temperature of ~210 K. Therefore, the critical temperature of NCCDW/CCDW phase transition is identified as ~150 and ~210 K for the cooling and heating processes, respectively. In order to precisely determine the transition temperature of NCCDW/CCDW phase, Raman frequencies of discernible peaks as a function of temperature are plotted in Fig. 4b, upon cooling/warming processes. Obviously, the number of vibration modes and their frequencies are dramatically changed at the transition temperature $T_c$ (marked by dashed lines), and this temperature is different between the cooling ($T_{c,cool} = 150$ K) and the warming ($T_{c,warm} = 210$ K) processes. The hysteresis temperature of $\Delta T = T_{c,warm} - T_{c,cool}$ $= 60$ K and the average transition temperature of $T_{c,avg} = (T_{c,warm} + T_{c,cool})/2 = 180$ K are then calculated accurately.

More temperature-dependent Raman spectra of 2H-TaS$_2$ with different thicknesses upon cooling/warming processes are presented in Supplementary Figs. 8 and 9. Interestingly, for the ~3 nm-thick 2H-TaS$_2$, we find a negligible variation for the discernible peaks under different temperature, suggesting that the CDW phase transitions of 2H-TaS$_2$ are strongly suppressed at this ultrathin thickness region (Supplementary Fig. 8). Temperature-dependent Raman frequencies of discernible peaks for 2H-TaS$_2$ of ~150 nm thick are also plotted in Fig. 4c, regarding cooling/warming processes. Contrastingly, Fig. 4d displays the hysteresis and average transition temperature plotted as a function of flake thickness. Apparently, $\Delta T$ increases with the reduction of sample thickness, while the $T_{c,avg}$ values are not changed substantially. As the film thickness decreases, the NCCDW/CCDW phase transition temperature decreases and vanishes at the critical thickness of ~3 nm, as presented in Fig. 4e. Shortly, such abovementioned results are in good agreement with

**Table 1 Comparison of the HER performances of 2H-TaS$_2$, 2H-MoS$_2$, and 1T-WS$_2$**

| Catalyst | Overpotential at 10 mA/cm$^2$ (mV vs RHE) | Tafel slope (mV/dec) | Exchange current density ($\mu$A/cm$^2$) | Ref. |
|---|---|---|---|---|
| 2H-MoS$_2$/Au(111) | 250 | 55–60 | 7.9 | 33 |
| 2H-MoS$_2$ bicontinuous network | 285 | 50 | 0.69 | 46 |
| Strained 1T-WS$_2$ nanosheets | 210 | 55 | 20 | 48 |
| Strained and vacant 2H-MoS$_2$ nanosheets | 170 | 60–98 | N/A | 34 |
| CVD 2H-MoS$_2$ nanosheets | 200 | 70 | 16.9 | 49 |
| 2H-TaS$_2$/Au foils | 65 | 33–42 | 100–179.47 | This work |

those of electrical transport measurements (temperature-dependent resistance) performed on exfoliated TaS$_2$[45], thus confirming the reliability of CVD-synthesized samples for detecting CDW phase transitions.

**Electrocatalytic performance of 2H-TaS$_2$.** A recent theoretical calculation predicted excellent electrocatalytic properties for metallic 2H-TaS$_2$ in HER, featuring high stability and active sites concentrated at the edges and in the basal-planes[33]. However, the direct application of 2H-TaS$_2$ in HER still remains unaddressed. Herein, the as-grown 2H-TaS$_2$ flakes on Au foils are directly used as electrocatalysts in HER, as schematically illustrated in Fig. 5a. In order to justify the catalytically active sites of 2H-TaS$_2$, density functional theory (DFT) calculations are firstly performed (Fig. 5b, c). The Gibbs free energy ($\Delta G_{H^*}$) is usually used to assess the catalytic performance, and a $\Delta G_{H^*}$ value close to zero usually indicates superior HER activity due to the optimal balance between absorption and removal of hydrogen atoms on the active sites[46]. In Fig. 5c, the values of $\Delta G_{H^*}$ for Ta-edge, S-edge, and basal-plane are calculated to be −0.04, −0.10, and 0.15 eV, respectively, which are all comparable with that of the MoS$_2$ edge but remarkably lower than that of the MoS$_2$ basal plane[46]. These calculated results indicate that the active sites of 2H-TaS$_2$ are concentrated both at the edges and in the basal-planes, in sharp contrast with that of MoS$_2$ that posseses inert surface catalytic properties[37]. It is worth mentioning that, Yakobson, B. I. et al. have demonstrated that the populated state of electrons near the lowest unoccupied state ($\xi_{LUS}$) is the key parameter of the adsorption strength of hydrogen on MX$_2$ surfaces. The basal-plane of 2H-TaS$_2$ possesses a relative low $\xi_{LUS}$ (<−5.8 eV), thus possessing relatively strong adsorption capability of hydrogen and thus enhanced catalytic activity[47].

Figure 5d displays the polarization curves of as-grown 2H-TaS$_2$ flakes with different thicknesses (all the tested samples show similar coverage of ~70%). The curves from Au foil and commercial Pt are also collected for comparison. Notably, at a cathodic current density ($j$) of 10 mA/cm$^2$, the overpotentials ($\eta$) of 2H-TaS$_2$ samples are falling in 65–150 mV, much lower than that of MoS$_2$ (170–250 mV)[33–36], possibly addressing the excellent HER activity of 2H-TaS$_2$. Furthermore, the linear portions of the Tafel plots in Fig. 5e are fitted to the Tafel equation ($\eta = b \log j + a$, where $j$ is the current density and $b$ is the Tafel slope), yielding Tafel slopes of 31, 33–42, and 110 mV/dec for Pt, 2H-TaS$_2$/Au, and Au foil, respectively. It is noteworthy that the Tafel slope (~33 mV/dec) for 2H-TaS$_2$ with the thickness of ~150 nm is very close to that of Pt (~31 mV/dec) and exceeds all the reported MX$_2$ candidates[36, 46, 48, 49]. The specific Tafel slope value should address a Volmer-Tafel mechanism for the HER of 2H-TaS$_2$.

By applying an extrapolation method to the Tafel plots, the exchange current density ($j_0$) is also obtained and displayed in Supplementary Fig. 10. A remarkable $j_0$ value of ~179.47 $\mu$A/cm$^2$ is achieved, which is superior to other MX$_2$ materials reported elsewhere[36, 46, 48, 49]. To address this, a comparison of the HER performances of CVD-derived 2H-TaS$_2$ and MX$_2$-based catalysts is displayed in Table 1. Particularly, after 5000 cycles, the 2H-TaS$_2$ flakes present much higher electrocatalytic activity than that of their initial states (Fig. 5f, and Supplementary Fig. 11). Notably, TaS$_2$ is a metallic TMDCs material, it is unstable under ambient condition and the surface can be oxidized, which results in extra low electrocatalytic activity for HER[50]. Through a facile HER cycling process, the surface oxides can be peeled off by hydrogen bubbles and the intrinsic electrocatalytic activities of 2H-TaS$_2$ are presented subsequently. Meanwhile, a microscopic morphology analysis reveals that the enhanced HER performance is closely correlated to the morphological evolution of 2H-TaS$_2$. The comparison of SEM and AFM morphologies of 2H-TaS$_2$ before and after 5000 cycles (Fig. 5f, and Supplementary Fig. 12) indicates that the flakes become thinner, smaller, and more disperse, but with invariable chemical composition of 2H-TaS$_2$ (Supplementary Fig. 13). In order to rule out the effect of the possible Pt contamination on the HER performance of 2H-TaS$_2$, we have re-measured the HER performance of 2H-TaS$_2$/Au by using the carbon rod as the counter electrode, and performed the Nafion proton exchange membrane assisted electrochemical measurement, respectively. Similar catalytic results have been achieved among the different methods, indicative of the high electrocatalytic performance of 2H-TaS$_2$/Au foils (Supplementary Fig. 14, and Supplementary Table 1). In our opinion, the cycling induced morphology change has three beneficial effects on the catalytic activity: 1) Shortening the interlayer electron-transfer pathways at a thinned domain; 2) increasing the active surface area by improving the accessibility of protons to basal-plane active sites; 3) increasing the density of active sites at the flake edge of 2H-TaS$_2$, considering that the $\Delta G_{H^*}$ values of both Ta-edge and S-edge are much closer to the thermo-neutral point than that of the basal-plane, and the edge sites are catalytically more active than that of the basal plane. Such conclusions are further confirmed by the electrochemical impedance spectra (ESI) (Supplementary Fig. 15), where a decrease in charge-transfer resistance is observed upon cycling. The extra low charge-transfer resistance (5–11 $\Omega$) in 2H-TaS$_2$/Au indicates the fast charge transfer between TaS$_2$ and Au (Fig. 5g, h). The effect of CVD synthesis temperature on the HER performance of TaS$_2$ was also presented in Supplementary Fig. 16, where low synthesis temperature (< 800 °C) has negligible effect on the HER performance of TaS$_2$, and the high synthesis temperature (> 800 °C) reduces the catalytic activity of TaS$_2$. This can be explained from the generation of different phases.

**Discussion**

In summary, we have developed facile LPCVD and APCVD routes for synthesizing large-area uniform, thickness controllable 2H-TaS$_2$ films and domains directly on Au foils, respectively. The high-quality 2D 2H-TaS$_2$ samples have proven to be attractive platforms for investigating fundamental physical phenomena (e.g.

CDW) associated with the dimensionality effect. More significantly, the metallic 2H-TaS$_2$ have been found to be an efficient electrocatalyst for the HER, even comparable to Pt, owing to its abundant active sites concentrated at edges and basal-planes, as well as the self-optimizing morphological change of 2H-TaS$_2$. We believe this work could be a significant advance towards the batch production and electrocatalytic applications of 2D metallic materials, and hope these results will motivate scientists to explore new efficient catalysts in the large materials family of MTMDCs for energy related applications.

## Methods

**Materials synthesis.** The TaCl$_5$ (Alfa Aesar, purity 99.5%) and S (Alfa Aesar, purity 99.5%) powders were used as precursors and the Au foil as the substrate (Alfa Aesar, purity 99.99%). All the sample growth was finished in a three-zone furnace (Lindberg/Blue M HTF55347c) equipped with a 1 inch diameter quartz tube. The temperature of Au foil, TaCl$_5$ and S powders were set at 750, 300 and 280 °C, respectively. Ar (100 sccm) and H$_2$ (10 sccm) were used as carrier gases. After a growth period, the furnace was opened, and the sample was cooled to room temperature in the flowing mixed gases of H$_2$/Ar (10/100 sccm).

**Materials characterization.** The samples were characterized by OM (Olympus BX51), SEM (Hitachi S-4800, 2 kV), XPS (Kratos Analytical AXIS-Ultra with monochromatic Al Kα X-ray), XRD (Shimadzu Thin Film, using Cu Kα radiation at room temperature in the 2θ range of 10 ~ 90°), Raman spectroscopy (Renishaw, Invia Reflex, excitation light of ~514 nm), TEM (JEOL JEM-2100F LaB6; acceleration voltage, 200 kV), and AFM (Dimension Icon, Bruker). The LEEM and μ-XPS elemental mapping data were acquired at the X-ray photoemission electron microscopy end station of the 09U (Dreamline) beamline of the Shanghai Synchrotron Radiation Facility. High resolution STEM-HAADF images were obtained on an aberration corrected transmission electron microscope JEM-ARM200F equipped with cold field emission gun with acceleration voltage of 200 kV.

**Electrochemical measurements.** All the electrochemical measurements were performed in a three-electrode system on CHI 760E electrochemical workstation (CH Instruments), using 2H-TaS$_2$/Au foil as the working electrode, a Pt foil or carbon rod as a counter electrode, and a saturated Ag/AgCl as a reference electrode. All the potentials were calibrated by a reversible hydrogen electrode (RHE). Linear sweep voltammetry with a scan rate of ~5 mV s$^{-1}$, from +0.10 to −0.70 V vs. RHE was conducted in 0.5 M H$_2$SO$_4$ (sparged with N$_2$, purity ~99.999%). The Nyquist plots were obtained with frequencies ranging from 100 kHz to 0.1 Hz at the overpotential of 10 mV. The impedance data were fitted to a simplified Randles circuit to extract the series and charge-transfer resistances.

**DFT calculations.** All theoretical calculations were performed within the framework of DFT using the Vienna ab initio simulation package (VASP)[51] with projector-augmented wave scheme. Then the Gibbs free energy for hydrogen adsorption, $\Delta G_{H^*}$, was estimated following the procedure described in a previous report[52].

**Data availability.** The data reported by this article are available from the corresponding author upon reasonable request.

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

## Acknowledgements

We gratefully acknowledge the financial support from the National Natural Science Foundation of China (Nos. 51290272, 51472008, 51472080, 51432002, 51520105003, 51522212, 51421002, 51672307, and 21673054), the Ministry of Science and Technology of China (Nos. 2016YFA0200103, 2016YFA0200700, 2013CB932603, and 2014CB921002), the Open Research Fund Program of the State Key Laboratory of Low Dimensional Quantum Physics (No. KF201601), the Strategic Priority Research Program of Chinese Academy of Sciences (No. XDB07030200), and the Key Research Program of Frontier Sciences, Chinese Academy of Sciences (Nos. QYZDB-SSW-JSC035, and QYZDB-SSW-SYS031).

## Author contributions

J.S. and X.W. contributed equally to this work. Y.Z. conceived and supervised the research project. J.S. developed and conducted the CVD growth of TaS$_2$, with Y.H., Z.Z., X.Z., M.H., and Q.F.'s assistance. Y.G. and L.G. performed the STEM-HAADF characterization. J.S., Y.H., Z.Z., X.Z., M.H., and Q.F. carried out the OM, XPS, XRD, SEM, AFM, TEM, LEEM and μ-XPS characterization. X.W., Y.H., and L.X. performed the electrochemical measurements. S.Z., Q.Z., and X.L. carried out the temperature-dependent Raman spectroscopy characterization. Y.L. performed the DFT calculations. All the authors discussed the results and commented on the manuscript.

## Additional information

**Competing interests:** The authors declare no competing financial interests.

