## [Peer Review File · Nature Communications]

Reviewers' comments:

Reviewer #1 (Remarks to the Author):

The authors developed the synthetic method of thickness- controllable TaS₂ through LPCVD and APCVD. The approach is meaningful because growth of metallic TMDC is very limited. Also, they showed TaS₂ which have extraordinary performance for hydrogen evolution reaction. However, several factors should be mentioned clearly.

1. The authors explained the flakes are getting smaller, thinner and more disperse after 5000 cycling of HER then the performance is enhanced.(Page 11, line 301) They claimed that the thinner effect brought short electron-pathway on basal plane and increasing surface area by improving the accessibility of proton to basal plane so HER performance is enhanced. But, smaller flakes also mean edge sites of TaS₂ would be increased, and the activity of HER can come from more edges than basal plane. I am not sure whether the good performance is from basal plane. The author need to be more careful and clear to explain about the active sites.

2. TaS₂ with 150 nm shows the best HER results in Figure 5d. However, the initial HER performance of figure 5f doesn't show as good as figure 5d, although they have same thickness (150 nm).The TaS₂ with 150 nm in Figure 5d is same as after 5000 cycling in figure 5f. That point is confusing. Does it have the best result only after cycling? If they have good results only after cycling, TaS₂ doesn't have good catalytic activities.

3. Did the author exclude Au foil effect on HER? Au could be evaporated during the growth then can be penetrated into TaS₂. The author didn't show any evidence about Au contamination.

4. From XPS, the reviewer didn't see any deconvolution peaks from Ta and S. TMDC can have various oxidation state therefore showing deconvolution is necessary.

Reviewer #2 (Remarks to the Author):

The paper by Shi et al reports the CVD growth of two-dimensional 2H TaS₂ on gold substrate and unusually high catalytic activities towards hydrogen evolution reactions, together with carefully structure characterizations using AFM, XPS, TEM and Raman. The paper reads very well and the results are new and interesting. In principle I agree the paper for acceptance of publication after minor revision.

1. I am pretty surprised to see so high HER activity from the 2H TaS₂. I noticed that the authors used Pt as the counter electrode in their electrochemical measurements. To rule out the possible Pt contamination of the 2H-TaS₂/Au foil working electrode, the authors are suggested to re-measure the HER performance by using carbon electrode as the counter electrode. The measurements will also be helpful to exclude the alternative explanation on electrochemical cycling enhanced electrocatalytic activity of 2H TaS₂ from Pt contamination.

2. Electrochemical impedance spectroscopy should offer experimental evidence to support the authors' claims on the effect of gold substrates on charge transfer and the kinetic origins of the high HER activities of 2H TaS₂. Suggest the authors add those data in the paper.

Reviewer #3 (Remarks to the Author):

The authors reported that a 2H-TaS₂/Au foil could be used as catalyst in high hydrogen evolution reaction. The manuscript is clearly written. But, there are somethings to be revised for readers.

1. The authors investigated the effect of temperature on Raman spectra. According to the Figure 4a, there is no significant change in behavior of Raman spectra between 80 to 210 K. It would be better to choose the critical temperature which indicates the effect of heating or cooling, significantly.
2. The DFT calculation is a powerful tool in atomic level calculations. In this work, the authors tried to justify the electrochemical reaction mechanism by using DFT calculation. They just calculated the Gibbs energy to show and explain the catalyst mechanism. It is not enough for readers. It would be better to bring at least a sentence about the catalytic performance and mechanism or make a reference.
3. What is the effect of temperature on catalytic performance?
4. In this paper, the authors mentioned the electrical transport, but they didn't about the amount of electrical transport. What is your definition from electrical transport? (e.g. mobility, conductivity, sheet resistance, current etc.)

In this revised version, we have made great efforts to address the points raised by reviewers by performing more supplementary experiments, especially for **Reviewer 2**, regarding the possible Pt contamination on the HER efficiency of 2H-TaS₂/Au foils sample with the Pt working electrode, and regarding the effect of synthesis temperature on the HER performance of TaS₂ raised by **Reviewer 3**.

Through these careful revisions, we sincerely hope that, our work would be of great interest to the broad readership of *Nature Communications* in both fundamental researches for the growth of metallic transition metal dichalcogenides and their applications in energy related fields.

Reviewer #1 (Remarks to the Author):

The authors developed the synthetic method of thickness-controllable TaS₂ through LPCVD and APCVD. The approach is meaningful because growth of metallic TMDC is very limited. Also, they showed TaS₂ which have extraordinary performance for hydrogen evolution reaction. However, several factors should be mentioned clearly.

Our response:

We are very grateful for the reviewer's very positive evaluation on the significance of our work, especially regarding the successful growth of metallic transition metal dichalcogenides (TMDCs) and the extraordinary performance for hydrogen evolution reaction (HER). The issues raised by the reviewer are considered very carefully and addressed point-by-point as follows.

1. The authors explained the flakes are getting smaller, thinner and more disperse after 5000 cycling of HER then the performance is enhanced (Page 11, line 301). They claimed that the thinner effect brought short electron-pathway on basal plane and increasing surface area by improving the accessibility of proton to basal plane so HER performance is enhanced. But, smaller flakes also mean edge sites of TaS₂ would be increased, and the activity of HER can come from more edges than basal plane. I am not sure whether the good performance is from basal plane. The author need to be more careful and clear to explain about the active sites.

Our response:

We are very thankful for the reviewer's very constructive comment. We agree with the reviewer that, the edge sites of TaS₂ will be increased for the smaller flakes after cycling. According to the theoretical calculation data shown in **Fig. 5c**, the free energies of H adsorption (ΔG_{H^*}) on Ta-edge (-0.04 eV) and S-edge (-0.10 eV) are much closer to thermos-neutral ($\Delta G_{H^*} = 0$ eV) than that of the basal-plane (0.15 eV). The edge sites should be more active in the electrocatalytic activity for HER than that of the basal plane. That is to say, the smaller TaS₂ flakes with more edge sites (after cycling) should possess much better HER performance than that of the initial larger TaS₂ domains.

According to the kind suggestion, we have clarified this discussion in **page 12** by "*In our opinion, the cycling induced morphology change has three beneficial effects on the catalytic activity..., 3) increasing the density of active site at the flake edge of 2H-TaS₂, considering that the ΔG_{H^*} values of both Ta-edge and S-edge are much closer to the thermos-neutral point than that of basal-plane, and the edge sites are catalytically more active than that of the basal plane...*"

2. TaS₂ with 150 nm shows the best HER results in Figure 5d. However, the initial HER performance of figure 5f doesn't show as good as figure 5d, although they have same thickness (150 nm). The TaS₂ with 150 nm in Figure 5d is same as after 5000 cycling in figure 5f. That point is confusing. Does it have the best result only after cycling? If they have good results only after cycling, TaS₂ doesn't have good catalytic activities.

Our response:

We are very grateful for the instructive comment raised by the reviewer. The HER results in **Figs. 5d,f** are derived from the similar TaS₂ samples with the same thickness. Notably, in consideration that the TaS₂ is a metallic TMDC material, which is unstable under ambient condition and the surface of TaS₂ can be oxidized at the atmospheric condition before the HER measurement. Ohsaka, T. *et al.*, has shown that, these surface oxides (TaO_x) have extra low electrocatalytic activity for HER, featured with much larger overpotential at 0.1 mA/cm⁻² (~250 mV) than that of TaS₂ (~1 mV) (Ohsaka, T. *et al.*, *J. Mater. Chem. A* 3, 16791–16800 (2015)). As a result, at the initial state, the 2H-TaS₂/Au has relatively low HER activity. Particularly, the surface oxides can be peeled off by hydrogen bubbles during the HER cycling process, and the intrinsic electrocatalytic activities of TaS₂ are presented subsequently (after 500 cycling). Therefore, it is proposed that, the TaS₂ flakes possess extra high electrocatalytic performance after removing the surface oxides through a facile HER cycling process.

We have added this discussion in **page 12** by “...*Notably, considering that the TaS₂ is a metallic TMDC material, which is unstable under ambient condition and the surface of TaS₂ can be oxidized at the atmospheric condition, and these surface oxides (TaO_x) have extra low electrocatalytic activity for HER⁵⁰. Through a facile HER cycling process, the surface oxides can be peeled off by hydrogen bubbles and the intrinsic electrocatalytic activities of 2H-TaS₂ are presented subsequently...*” We have also added this reference in **page 12** (in **Ref. 50**).

3. Did the author exclude Au foil effect on HER? Au could be evaporated during the growth then can be penetrated into TaS₂. The author didn't show any evidence about Au contamination.

Our response:

We are very thankful for the reviewer's very constructive comment. The polarization curves in **Fig. 5d** have shown that the Au foil has very low electrocatalytic activity.

In order to further exclude the possible penetration of Au atoms into the TaS₂ layers in the CVD growth process, we have performed XPS measurements of transferred TaS₂ on SiO₂/Si (**Fig. R1**, also shown in **Supplementary Fig. 6** in the revised manuscript). The Au 4f characteristic peaks are not observed, indicating that the Au element is not penetrated into TaS₂ during the CVD growth process.

We have added this discussion in **page 6** by “...*The XPS measurement of transferred 2H-TaS₂ on SiO₂/Si was also performed to exclude the possible Au penetration into the 2H-TaS₂ layers during the CVD growth process (Supplementary Fig. 6)...*”

More details were provided in Supplementary Information (**Supplementary Fig. 6**) by “*The XPS measurement of transferred 2H-TaS₂ flakes on SiO₂/Si was also performed to exclude the possible mixing of Au in the 2H-TaS₂ layers during the CVD growth process (Supplementary Fig. 6). The Au 4f characteristic peaks are not observed, strongly indicating that Au is not penetrated into TaS₂ layers during the growth process.*”

Figure R1 (also shown in Supplementary Fig. 6 in the revised manuscript) | XPS spectrum of transferred 2H-TaS₂ on SiO₂/Si acquired over a wide range of binding energies (0-1100 eV). Inset is the zoom-in scan from the Au 4f range.

4. From XPS, the reviewer didn't see any deconvolution peaks from Ta and S. TMDC can have various oxidation state therefore showing deconvolution is necessary.

Our response:

We are very thankful for the reviewer's very constructive comment. We have added the deconvolution peaks from Ta and S in Fig. R2 (also shown in Fig. 1b and Fig. 2b). We have also supplied a sentence in page 4 by "...Notably, additional peaks at 28.4 and 26.9 eV are attributed to Ta⁵⁺, in consideration of the oxidation susceptibility of metallic TaS₂..."

Figure R2 (also shown in Fig. 1b and Fig. 2b in the revised manuscript) | XPS peaks for Ta and S from LPCVD and APCVD derived-TaS₂ on Au foils.

Reviewer #2 (Remarks to the Author):

The paper by Shi et al reports the CVD growth of two-dimensional 2H TaS₂ on gold substrate and unusually high catalytic activities towards hydrogen evolution reactions, together with carefully structure characterizations using AFM, XPS, TEM and Raman. The paper reads very well and the results are new and interesting. In principle I agree the paper for acceptance of publication after minor revision.

Our response:

We are very grateful for the reviewer's very positive evaluation on the novelty of our work. The issues raised by the reviewer are considered very carefully and addressed point-by-point as follows.

1. I am pretty surprised to see so high HER activity from the 2H TaS₂. I noticed that the authors used Pt as the counter electrode in their electrochemical measurements. To rule out the possible Pt contamination of the 2H-TaS₂/Au foil working electrode, the authors are suggested to re-measure the HER performance by using carbon electrode as the counter electrode. The measurements will also be helpful to exclude the alternative explanation on electrochemical cycling enhanced electrocatalytic activity of 2H TaS₂ from Pt contamination.

Our response:

We are very thankful for the reviewer's very constructive comment.

We have re-measured the HER performance of 2H-TaS₂ by using the carbon rod as the counter electrode, and performed the Nafion proton exchange membrane assisted electrochemical measurements. Notably, the proton exchange membrane only permits the transfer of protons but impedes the other species, this method has been widely employed in electrocatalytic HER and fuel cells (E. Schaak, R. *et al.*, *J. Am. Chem. Soc.* 135, 9267–9270 (2013); Nørskov, J. K. *et al.*, *J. Am. Chem. Soc.* 127, 5308–5309 (2005); Friedrich, K. A. *et al.*, *Angew. Chem. Int. Edit.* 128, 752–756 (2016); Shan, Y. *et al.*, *Nano Lett.* 4, 345–348 (2016)).

Three results have been presented to rule out the possible Pt contamination on the HER performance of 2H-TaS₂:

(1) The HER performances of 2H-TaS₂/Au samples with different thicknesses have been re-measured by using the carbon rod as the counter electrode, with the results shown in **Figs. R3a–c** (also shown in **Supplementary Figs. 14a–c**, and **Supplementary Table 1** in the revised manuscript). At the cathodic current density (j) of 10 mA/cm², the overpotentials (η) are falling in 100–190 mV, the Tafel slopes are calculated to be 50–55 mV/dec, and the exchange current density (j_0) are measured to be 85–143.2 μ A/cm². Such three parameters are very close to the results achieved by using Pt as the counter electrode, preliminary indicating that the Pt counter electrode has very small effect on the HER performance of 2H-TaS₂. Notably, in consideration of the relatively low conductivity of carbon rod, the Tafel slope values are slightly larger than that of the results obtained from the HER measurement using Pt counter electrode;

(2) The Nafion proton exchange membrane assisted electrochemical measurements are also performed, as shown in **Figs. R3d–f** (also shown in **Supplementary Figs. 14d–f**, and **Supplementary Table 1** in the revised manuscript). The overpotentials at $j = 10$ mA/cm² are 80–160 mV, the Tafel slopes are 45–49 mV/dec, and the j_0 values are 90–149.5 μ A/cm², such results are very close to those of Pt as the counter electrode, further suggestive the negligible effect of Pt on the HER performance of TaS₂/Au;

(3) The XPS characterizations of 2H-TaS₂ after HER measurements were also performed (with the data shown in **Supplementary Fig. 13**). The Pt characteristic peaks are not observed, indicating that Pt is not covered on or mixed in TaS₂ during the HER process.

The aforementioned three results suggest that, the Pt counter electrode has negligible effect on the HER performance of 2H-TaS₂.

We have added this discussion in **page 12** by “...*In order to rule out the effect of the possible Pt contamination on the HER performance of 2H-TaS₂, we have re-measured the HER performance of 2H-TaS₂/Au by using the carbon rod as the counter electrode, and performed the Nafion proton exchange membrane assisted electrochemical measurement, respectively. Similar catalytic results have been achieved among the different methods, highly indicative the extra high electrocatalytic performance of 2H-TaS₂/Au foils (Supplementary Fig. 14, and Supplementary Table 1).*...”

More results were provided in Supplementary Information (**Supplementary Fig. 14**) by “*In order to rule out the possible effect of Pt contamination, we have re-measured the HER performance of various 2H-TaS₂ samples on Au foils by using the carbon rod as the counter electrode. The overpotentials at the cathodic current density of 10 mA/cm² are falling in the range of 100–190 mV, and the Tafel slope values are in the range of 50–55 mV/dec. Such results approach to those obtained from the Pt counter electrode based HER measurements. Additionally, by applying extrapolation method to the Tafel plots, the exchange current densities (j_0) are also achieved in the range of 85–143.2 $\mu\text{A}/\text{cm}^2$, similar with those of using the Pt counter electrode (Supplementary Figs. 14a–c, and Supplementary Table 1). Notably, such three parameters are very close to the results derived by using Pt as the counter electrode. Briefly, the Pt counter electrode has very small effect on the HER performance of 2H-TaS₂. And the slightly larger Tafel slope value than that of using the Pt counter electrode based HER measurement may be induced by the relatively low conductivity of the carbon rod.*”

We also performed the Nafion proton exchange membrane assisted electrochemical measurements by using Pt counter electrode. This is because the proton exchange membrane only permits the transfer of protons but impedes the other species. The overpotentials for the different samples at $j = 10 \text{ mA}/\text{cm}^2$ are in the range of 80–160 mV, the Tafel slopes are 45–49 mV/dec, and the j_0 values are 90–149.5 $\mu\text{A}/\text{cm}^2$, such values are very close to the results only by using Pt as the counter electrode (Supplementary Figs. 14d–f, and Supplementary Table 1). This result again suggests the negligible effect of Pt contamination on the HER performance of TaS₂.”

Figure R3 (also shown in Supplementary Fig. 14 in the revised manuscript) | Additional HER measurements of as-grown 2H-TaS₂ by using carbon rod as the counter electrode, and by using the Nafion proton exchange membrane assisted electrochemical measurements (with Pt counter electrode), respectively. (a–c) Polarization curves (iR-corrected), corresponding Tafel plots, and calculated exchange current densities of as-grown 2H-TaS₂ with different thicknesses by using carbon rod as the counter electrode. (d–f) Polarization curves (iR-corrected), corresponding Tafel plots, and calculated exchange current densities of as-grown 2H-TaS₂ with different thicknesses achieved by Nafion proton exchange membrane assisted electrochemical measurements.

Table 1. Comparison of the HER performances of 2H-TaS₂ by using different test methods

Method	Overpotential at 10 mA/cm ² (mV vs RHE)	Tafel slope (mV/dec)	Exchange current density (μA/cm ²)
Pt counter electrode	65–150	33–42	100–179.47
carbon rod counter electrode	100–190	50–55	85–143.2
Nafion proton exchange membrane	80–160	45–49	90–149.5

2. Electrochemical impedance spectroscopy should offer experimental evidence to support the authors' claims on the effect of gold substrates on charge transfer and the kinetic origins of the high HER activities of 2H TaS₂. Suggest the authors add those data in the paper.

Our response:

We are very thankful for the reviewer's very kind suggestion. The electrochemical impedance spectroscopies (ESI) of 2H-TaS₂ samples and Au foils have been added in Fig. R4 (also shown in Figs. 5g,h in the revised manuscript). We have also added some discussion in page 13 by "...The extra low charge-transfer resistance (5–11 Ω) in 2H-TaS₂/Au indicates the fast charge transfer between TaS₂ and Au (Figs. 5g,h)..."

Figure R4 (also shown in Figs. 5g,h in the revised manuscript) | Electrochemical impedance spectroscopies of 2H-TaS₂ flakes with different thicknesses, as well as the Au foil substrate.

Reviewer #3 (Remarks to the Author):

The authors reported that a 2H-TaS₂/Au foil could be used as catalyst in high hydrogen evolution reaction. The manuscript is clearly written. But, there are some things to be revised for readers.

Our response:

We are very grateful for the reviewer's very positive evaluation on our work. The issues raised by the reviewer are considered very carefully and addressed point-by-point as follows.

1. The authors investigated the effect of temperature on Raman spectra. According to the Figure 4a, there is no significant change in behavior of Raman spectra between 80 to 210 K. It would be better to choose the critical temperature which indicates the effect of heating or cooling, significantly.

Our response:

We are very thankful for the reviewer's kind comment. For the cooling process, a broad Raman peak is observed below 100 cm⁻¹ at >150 K. In contrast, the fine peak (79.8 cm⁻¹, indicated by a red arrow in **Fig. 4a**) is visible at <150 K, suggesting that a NCCDW/CCDW phase transition takes place at ~150 K. Notably, a similar change tendency is also recorded for the heating process, namely, a broad Raman peak is visible below 100 cm⁻¹ at >210 K and the fine peak is noticeable at <210 K, indicating that the NCCDW/CCDW phase transition takes place at ~210 K for the heating process.

Briefly, the critical temperature is defined as ~150 and ~210 K for the cooling and heating processes, respectively. A similar result for the mechanically exfoliated 1T-TaS₂ sample was also reported by Tsen, A. W. *et al.*, (Ref. 44, Tsen, A. W. *et al.*, *Phys. Rev. B* 94, 201108 (R) (2016)).

In order to precisely determine the critical temperature of NCCDW/CCDW phase transition, we have re-measured the Raman spectra of 2H-TaS₂ flakes from 80 to 210 K in **Fig. 4a** and added this discussion in **page 10** by "...Therefore, the critical temperature of NCCDW/CCDW phase transition is identified as ~150 and ~210 K for the

cooling and heating processes, respectively...”

2. The DFT calculation is a powerful tool in atomic level calculations. In this work, the authors tried to justify the electrochemical reaction mechanism by using DFT calculation. They just calculated the Gibbs energy to show and explain the catalyst mechanism. It is not enough for readers. It would be better to bring at least a sentence about the catalytic performance and mechanism or make a reference.

Our response:

We are very grateful for the constructive comment raised by the reviewer. We agree with the reviewer that, the DFT calculation is a powerful tool to understand and explain the catalyst mechanism.

We have supplemented some discussion about the catalytic performance and the internal mechanism of 2H-TaS₂ according to the DFT calculation results from the published references in **page 11** by “...*In Fig. 5c, the values of ΔG_{H^*} for Ta-edge, S-edge, and basal-plane are calculated to be -0.04 , -0.10 , and 0.15 eV, respectively, which are all comparable with that of the MoS₂ edge but remarkably lower than that of the MoS₂ basal plane⁴⁶. These calculated results indicate that, the active sites of 2H-TaS₂ concentrated both at the edges and in the basal-planes, in sharp contrast with that of MoS₂ owing inert surface catalytic property³⁷. It is worthy of mentioning that, Yakobson, B. I. et al. have demonstrated that, the populated state of electrons near the lowest unoccupied state (ζ_{LUS}) is the key parameter of the adsorption strength of hydrogen on MX₂ surfaces. The basal-plane of 2H-TaS₂ possesses a relative low ζ_{LUS} (<-5.8 eV), thus possessing relatively strong adsorption capability of hydrogen and thus enhanced catalytic activity⁴⁷ ...” We have also added this reference in **page 11** (in **Ref. 47**).*

3. What is the effect of temperature on catalytic performance?

Our response:

We are very thankful for the reviewer’s constructive comment. We have measured the HER performance of TaS₂ flakes with the similar coverage and domain size but synthesized at different temperature (650, 700, 750, 800, and 850 °C) in **Figs. R5a,b** (also shown in **Supplementary Figs. 16a,b** in the revised manuscript). Interestingly, the HER performances of TaS₂ flakes synthesized at 650, 700, and 750 °C are very similar, featured with the similar overpotential values at the cathodic current density of 10 mA/cm² of 120–125 mV, and the Tafel slope values of 45–48 mV/dec. This phenomenon indicates that, the growth temperature (from 650 to 750 °C) has negligible effect on the HER performance of TaS₂. However, for the TaS₂ flakes synthesized at higher temperature (800 and 850 °C), the overpotential (at the cathodic current density of 10 mA/cm²) and Tafel slope values are raised to 152–171 mV and 58–62 mV/dec, respectively. This indicates that the high synthesis temperature reduces the HER performance of the derived TaS₂ flakes.

Liu, Z., and Yakobson, B. I. et al. have demonstrated that, 1T-TaS₂ is dominant for the high temperature (900 °C) synthesized sample (**Refs. 32,47**), and the subsequent DFT calculation reveals that 1T-TaS₂ has relatively low catalytic activity (**Ref. 37**). In this regard, we speculate that, the high temperature (800 and 850 °C) synthesized samples contain 1T-TaS₂, which should address their reduced HER performances. The Raman spectra of transferred TaS₂ (synthesized at 800 and 850 °C) on SiO₂/Si are also collected to determine the phase state of TaS₂ in **Figs. R5c,d** (also shown in **Supplementary Figs. 16c,d** in the revised manuscript). Notably, the Raman characteristic peaks of 2H-TaS₂ (E_{2g} ~ 286.4 cm⁻¹ and A_{1g} ~ 401.4 cm⁻¹) and 1T-TaS₂ (E_{2g} ~ 244.2 cm⁻¹ and A_{1g} ~ 396.5 cm⁻¹) are

observed concurrently, confirming that the 1T- and 2H-TaS₂ are coexisting for the high temperature (800 and 850 °C) synthesized samples.

Briefly, the low synthesis temperature (<800 °C) has very little effect on the HER performance, and the catalytic activity of TaS₂ can be reduced at higher growth temperature (>800 °C). This difference is mainly mediated by their different phases of 2H-TaS₂ and 1T-TaS₂.

Figure R5 (also shown in Supplementary Fig. 16 in the revised manuscript) | HER measurements of TaS₂ synthesized at different temperature (650, 700, 750, 800, and 850 °C). (a,b) Polarization curves (iR-corrected) and corresponding Tafel plots of TaS₂ synthesized at different temperature (650, 700, 750, 800, and 850 °C). (c,d) Raman spectra of TaS₂ synthesized at different temperature (800 and 850 °C).

We have supplied this details in Supplementary Information (Supplementary Fig. 16) by “The HER performances of TaS₂ samples synthesized at different temperatures were also measured to detect the effect of synthesis temperature on the catalytic activity (Supplementary Figs. 16a,b). In order to exclude the influence of the edge density difference on the hydrogen evolution rate, the TaS₂ samples were selected to possess the similar coverage (~70%) and domain size (edge length of ~5 μm). Notably, the HER performances of TaS₂ synthesized at 650, 700, and 750 °C are very similar, featured with the overpotential values (at the cathodic current density (j) of 10 mA/cm²) of 120–125 mV, and the Tafel slope values of 45–48 mV/dec. This phenomenon indicates that, the growth temperature (from 650 to 750 °C) has negligible effect on the HER performance of TaS₂.

However, for the TaS₂ flakes synthesized at high temperature (800 and 850 °C), the overpotential (j = 10 mA/cm²) and Tafel slope values are raised to 152–171 mV and 58–62 mV/dec, respectively. Liu, Z., and Yakobson, B. I. et al. have demonstrated that, 1T-TaS₂ is dominant for the high temperature (900 °C) synthesized sample^{2,12}. Corresponding DFT calculation reveals that, 1T-TaS₂ has relatively low catalytic activity¹³. Herein, we speculate that the high temperature (800 and 850 °C) synthesized samples contain 1T-TaS₂, resulting in their reduced HER performance.

For more proof, the Raman spectra of transferred TaS₂ (synthesized at 800 and 850 °C) on SiO₂/Si are also

collected to determine the phase state, as shown in Supplementary Figs. 16c,d. Notably, the Raman characteristic peaks of 2H-TaS₂ (E_{2g} ~286.4 cm⁻¹ and A_{1g} ~401.4 cm⁻¹) and 1T-TaS₂ (E_{2g} ~244.2 cm⁻¹ and A_{1g} ~396.5 cm⁻¹) are observed concurrently, confirming that 1T- and 2H-TaS₂ are coexisting for the high temperature (800 and 850 °C) synthesized samples. Briefly, the low synthesis temperature (<800 °C) has very little effect on the HER performance of the CVD-derived TaS₂, and the high synthesis temperature (>800 °C) reduces the catalytic activity of TaS₂. This can be explained from the generation of different phases.”

We have also added a short discussion in **page 13** by “...*The effect of CVD synthesis temperature on the HER performance of TaS₂ was also presented in Supplementary Fig. 16, where low synthesis temperature (<800 °C) has negligible effect on the HER performance of TaS₂, and the high synthesis temperature (>800 °C) reduces the catalytic activity of TaS₂. This can be explained from the generation of different phases...*”

4. In this paper, the authors mentioned the electrical transport, but they didn't about the amount of electrical transport. What is your definition from electrical transport? (e.g. mobility, conductivity, sheet resistance, current etc.)

Our response:

We are very grateful for the reviewer's very constructive comment. The electrical transport in the revised manuscript has been defined in **page 6** by “...*Recent electrical transport measurements (temperature-dependence resistance) have revealed that...*”

REVIEWERS' COMMENTS:

Reviewer #1 (Remarks to the Author):

The authors demonstrated the synthetic method of metallic TMD materials. In this field, the growing metallic TMD is limited because it is very difficult and get oxidized easily. However, the authors developed the method and did well characterization.

1. The reviewer asked them to clarify the active sites for HER. They added the explanation of the morphology change during the cycles in manuscript and free energy calculation that they mentioned can support that the edge sites are more active than basal plane.
2. The reviewer understood what they explained.
3. The authors proved that there is no Au contamination through XPS.
4. The reviewer asked the authors to deconvolute XPS. Although Ta⁵⁺ is observed from the data, they mentioned the surface of oxidation in manuscript.

Reviewer #2 (Remarks to the Author):

The authors have addressed all my in the revised manuscript. The paper is ready for publication in Nature Comm.

Reviewer #3 (Remarks to the Author):

It seems that the manuscript is well revised by following the reviewers' comments. So, I think that the manuscript is good enough to be published in Nature Communications.

REVIEWERS' COMMENTS:

Reviewer #1 (Remarks to the Author):

The authors demonstrated the synthetic method of metallic TMD materials. In this field, the growing metallic TMD is limited because it is very difficult and get oxidized easily. However, the authors developed the method and did well characterization.

1. The reviewer asked them to clarify the active sites for HER. They added the explanation of the morphology change during the cycles in manuscript and free energy calculation that they mentioned can support that the edge sites are more active than basal plane.

2. The reviewer understood what they explained.

3. The authors proved that there is no Au contamination through XPS.

4. The reviewer asked the authors to deconvolute XPS. Although Ta^{5+} is observed from the data, they mentioned the surface of oxidation in manuscript.

Our response:

We are very grateful for the reviewer's very positive evaluation on the significance of our work, especially regarding the growth of metallic transition metal dichalcogenides. We are also very thankful for the reviewer's admission about our response.

Reviewer #2 (Remarks to the Author):

The authors have addressed all my in the revised manuscript. The paper is ready for publication in Nature Comm.

Our response:

We are very thankful for the reviewer's agreement for publication our manuscript in *Nature Communications*.

Reviewer #3 (Remarks to the Author):

It seems that the manuscript is well revised by following the reviewers' comments. So, I think that the manuscript is good enough to be published in Nature Communications.

We are very thankful for the reviewer's agreement for publication our manuscript in *Nature Communications*.